# Media and Institutional Investors Focus on the Impact on Corporate Sustainability Performance

**Chuanzhe Liu and Xu Wang ***

School of Management and Economy, China University of Mining and Technology, Xuzhou 221116, China
* Correspondence: wx1424457643@163.com

**Abstract:** This paper takes China's A—share listed companies from 2010 to 2020 as a sample to study the impact of media attention, institutional investor attention, and their interaction on the sustainable development performance of enterprises as well as the regulatory role of internal control levels. The research shows that the attention of the media and institutional investors has a significant improvement effect on the sustainable development performance of enterprises; there is a significant substitution when they play a role at the same time. This substitution effect is significantly affected by the level of the internal control of enterprises, that is, with the improvement in the level of the internal control of enterprises, the substitution effect between the two will be weakened, and ultimately positively affect the level of the sustainable development of enterprises, which provides an effective policy basis for the sustainable development of listed enterprises and the use of external supervision by the government to supervise listed enterprises.

**Keywords:** media attention; institutional investors; substitution effect; internal control (IC); corporate sustainable development

## 1. Introduction

In recent years, environmental problems have gradually become a major problem restricting the healthy development of China's economy, and the contradiction between environmental protection and economic development has become more and more serious. In order to promote the green and healthy development of the economy and society, President Xi Jinping announced, at the 75th session of the United Nations General Assembly, that China will adopt a "two-step" strategy to achieve "peak carbon" by 2030 and "carbon neutrality" by 2060. As an indispensable part of economic development, how to effectively supervise and improve the level of sustainable development of enterprises while achieving green development has become an urgent problem to be solved by academia.

With the rapid development of informatization, media supervision has gradually become an important means to effectively make up for the lack of development of the legal environment in emerging markets. The media can enhance the exposure of enterprises, promote enterprises to perform corporate environmental protection business in a way encouraged by positive behavior and national policies and regulations, and improve their environmental performance. Meanwhile, the media, as an integral part of social supervision and an external governance force [1,2], can improve information transparency by transmitting, processing, mining, and disclosing information to alleviate the problem of information asymmetry in the capital market [3], reduce the impact of information asymmetry on corporate development, enhance the stakeholders' confidence in the company, facilitate stakeholders to make correct decisions, and enhance corporate value. In addition, compared with small- and medium-sized shareholders, investors in listed companies often have information and capital advantages [4,5]. Institutional investors have diversified information channels, rich management experience, and professional analyst teams, which can effectively search, screen, and process information, also enabling institutional investors

to effectively participate in corporate governance and supervision and discover early problems in the operation of enterprises and urge managers to make corrections. The capital advantage enables institutional investors to have economies of scale when monitoring the behavior of the company's management, which affects the flow of funds in the capital market to a certain extent and puts pressure on listed companies. These two advantages prompt institutional investors to intervene to improve corporate governance and increase enterprise value. Environmental protection has increasingly become an important factor for companies to influence their survival and development, and the increasing importance of environmental performance in business management can be found in the setting of green thresholds for public financing and bank credit, and the shift in consumer preferences for green and safe product demand. It has been shown that the improvement in environmental performance can enhance the competitive advantage and improve the long-term financial performance of enterprises [6]. A decline in environmental performance may lead to a crisis of corporate legitimacy and penalties from the government and the public, which is detrimental to the long-term development of the company. This makes institutional investors with higher shareholdings more concerned about the environmental issues of the firm [7]. However, existing studies have mainly focused on the impact of media and institutional investors on corporate social responsibility [8,9], or on corporate financial performance, and have mainly concentrated on the impact of a single external monitoring mechanism on corporate development, lacking research on the level of corporate sustainability. Furthermore, the media and institutional investors work together as external monitoring mechanisms, and the monitoring effect of both on listed companies cannot be completely separated. However, what is the effect between media attention and institutional investors when they play a joint monitoring role? Existing studies do not provide an answer.

To address the above questions, this paper focused on the following studies. First, this paper examined whether there was a positive relationship between the media and institutional investors and corporate sustainability performance. Second, this paper examined whether media attention and the shareholding of institutional investors together play a "complementary" or "substitution" role in monitoring corporate sustainable development. Third, internal control as an internal management system, pathway, and intermediary mechanism to achieve corporate governance goals is an important way to influence the level of corporate governance. Therefore, this paper examined the interaction between the two as well as the impact of the level of internal control on the utility.

The marginal contributions of this paper are as follows. First, it breaks through the limitation that most studies have only analyzed the supervisory role of one subject, and explores the possible complementary or alternative roles of different external supervisory instruments, which enriches the study of the external supervisory instruments of enterprises. Second, it examines the relationship between the level of internal control of enterprises and the interaction between the two, which increases the applicability of the research findings. Third, it adopts environmental performance and financial performance to comprehensively measure the sustainable development performance of enterprises, and analyzes the realization path of sustainable development from the micro level, which has strong practical significance for promoting sustainable development.

The rest of this paper is organized as follows. Section 2 provides a literature review of relevant studies. Section 3 presents the theoretical analysis and the research hypothesis. Section 4 defines the relevant indicators and constructs the empirical model. Section 5 discusses the empirical results. Section 6 summarizes the research findings and makes policy recommendations.

## 2. Literature Review

### 2.1. A Review of Relevant Research on Media Attention

With the rapid development of modern technology, the media is playing an increasingly important role in social and economic life. News media, mainly newspapers and news, can quickly and comprehensively transmit information to the public. Media reports allevi-

ate the information asymmetry between enterprises and the public [10], which is beneficial for enterprises to establish a good image, which in turn brings positive economic effects.

The media influences the level of corporate sustainability in two main ways. First, the impact of media coverage on the financial performance of firms. Timely and extensive media coverage reduces the degree of information asymmetry between companies and external investors, which facilitates and influences the judgments and decisions of the stakeholders, thus affecting the economic performance of companies [11]. In turn, the media will pay more attention to companies with high performance levels and raise their visibility, which in turn will bring more public attention and monitoring behavior [12]. Luo et al. [13] studied the effect of different types of coverage on firm performance and showed a positive relationship between positive media coverage and firm performance and the opposite for negative coverage. Yuan and Xiong [14] found a negative moderating effect of media attention on the effect of ESG on firm performance, suggesting that in the case of excessive media attention, it may trigger a counterproductive effect and weaken the moderating effect of ESG performance on firm performance. Tao and Jin [15] found a mediating effect of media attention in the interaction between social responsibility information disclosure and corporate financial performance. The enhancement of corporate value by media attention is also reflected in the impact on the stock value. Higher media coverage of corporate social responsibility behavior enhances the corporate value and stock returns, and the visibility of the behavior has a more pronounced effect on the corporate value for companies that are actively involved in social responsibility activities [16], and stocks with high media attention are more likely to receive consistently high levels of investor attention, which leads to higher returns over time [17]. In addition, Kang et al. [18] found that media coverage showed a significant negative relationship with the risk of stock price collapse. Not only that, media attention can also alleviate corporate financing constraints [19], reduce corporate equity and debt financing costs [20], indirectly reduce corporate expenses, and improve corporate performance. Second, the impact of media attention on corporate environmental performance. On one hand, the strong public pressure of media attention can monitor and regulate corporate environmental mismanagement, and companies have an incentive to justify environmental legitimacy by increasing their environmental capital expenditure pathways [21,22]. On the other hand, the strong public pressure of media attention will compress the space and time for the self-interested behavior of corporate managers and help to improve the corporate environmental performance by raising awareness of the environmental legitimacy process and using increased environmental capital expenditure as a way to send signals of legitimate management in response to environmental legitimacy pressures [23,24]. At the same time, the media, as an information intermediary for interest stakeholders, can positively motivate companies to fulfill their environmental responsibility and enhance their competitiveness by their high level of interest in companies [25,26].

### 2.2. A Summary of Relevant Research on Institutional Investors' Shareholdings

As an important force in the capital market, institutional shareholders can take advantage of their capital and information to timely identify problems in corporate development and participate in corporate governance. Institutional investors participate in corporate governance in two main ways. First, institutional investors can enhance the corporate value. Institutional investors, as external supervisors of listed companies, influence the information environment of listed companies and improve the information transparency of companies [27], thus reducing the degree of information asymmetry, stabilizing the company share prices [28,29], and enhancing stock returns [30]. Ward et al. [31] conducted a detailed study on the behavior of institutional investors and found that institutional investors focused on firms that had invested in firms with high value ranking, became more involved in their governance, and gained revenue while enhancing the value of the firm. Clay [32] further found that the positive relationship between institutional investor ownership and firm value was more significant in firms with high free cash flows. Shi and Tong [33] studied the relationship between institutional investors and company value

during the post-share reform period, and the results showed that there was a significant positive correlation between the shareholding ratio of institutional investors and company value, which promoted the improvement in the company value. Zhao and Guo [34] found differences in the effects of different types of institutional investors on firm value, where stress-sensitive institutional investors' holdings had insignificant effects on firm value, while stress-resistant institutional investors significantly increased the firm value. Second, Institutional investors are more inclined to participate proactively in corporate governance [35,36] and are concerned about the sustainability and long-term prospects of the company [37]. Institutional investors judge whether a company is worthy of investment based on the environmental information disclosed by the company [38,39], and a high level of corporate disclosure has a higher profitability and social status [40]. Institutional investors have information advantages because they have diversified information channels, rich management experience, and a professional team of analysts who can effectively search, screen, and process information [41], which also enables institutional investors to obtain the environmental information of enterprises in a timely manner, identify the environmental problems of enterprises early, and urge managers to make corrections [42].Wang et al. [43] found that institutional investors have the ability and motivation to influence corporate decision making based on their own traits and strengths, and tend to support strategic decisions of corporate social responsibility. Li and Lu [7] found that institutional investors with higher shareholdings were more concerned about the environmental issues of companies and emphasized sustainability and long-term performance. Guan and Que [44] explored the relationship between performance feedback, institutional investor shareholding, and corporate environmental performance, and showed that institutional investor shareholding played a moderating role in the relationship between expectation surplus and environmental performance. As the percentage of institutional investor shareholding increased, the more institutional investors were able to play a role in corporate governance and monitoring, reducing the negative impact of expectation surplus on environmental performance.

To sum up, there are many studies on the impact of media and institutional investors on corporate financial and environmental performance indicators, but there have been few studies on the impact on corporate sustainable development performance. In addition, are the two supervisors "promoting" or "weakening" each other in giving play to the role of corporate governance? Furthermore, as an extremely important factor affecting the decision-making ability of enterprises, can the level of internal control be used as an important factor to regulate the media and institutions to play a supervisory role in corporate governance?

In order to answer the above questions, this paper selected A-share listed companies on the Shanghai and Shenzhen stock markets from 2010 to 2020 as a sample to explore the effect of the media and institutional investors' shareholding on the corporate sustainable development performance as well as the interaction between them, and further explore whether this effect will be affected by the level of internal control.

## 3. Theoretical Analysis and Research Hypotheses

### 3.1. Media Attention, Institutional Investor Shareholding, and Corporate Sustainability Performance

As an effective monitoring mechanism, media attention can affect the sustainable development performance of enterprises through various channels. First, the timeliness and extensiveness of media reports reduces the degree of information asymmetry between enterprises and investors, which is beneficial for stakeholders to make judgments and decisions, and affects the enterprise value and economic performance [45]. Second, negative media reports can put companies under reputational pressure, and deep and serious media negative reports can lead to strong negative market reactions and damage corporate performance [46]. In addition, the media, as a third-party supervisor, is one of the important driving forces for enterprises to actively fulfill their social responsibilities [47]. The media obtains relevant information through newspapers, the Internet, and other media, and indirectly exerts pressure on enterprises through public opinion. In order to maintain their good



image, enterprises will actively fulfill their social responsibilities and respond positively to media reports [1]. Therefore, media attention can reduce information asymmetry and form a certain external pressure on enterprises to indirectly participate in corporate governance and improve the sustainable development performance of enterprises. Accordingly, this paper proposes Hypothesis 1:

**Hypothesis 1 (H1).** *The higher the media attention, the higher the level of corporate sustainability performance.*

In addition to media reports, institutional investors can also act as an effective supervisory body to influence the sustainable development performance of listed companies. Institutional investors can effectively improve the sustainable development performance of enterprises with the help of their own capital and information advantages. Specifically, on one hand, institutional investors can increase their voting rights by soliciting proxy rights or shareholder proposals, etc., which can affect corporate management decisions to a certain extent, alleviate the problem of the insufficient supervision ability of small and medium shareholders, and participate in corporate governance [48]. On the other hand, institutional investors can take advantage of the information, professional, and talent advantages of their major shareholders to effectively supervise the management of the company, and such effective supervision can increase the value of the company. Institutional investors can benefit from this. The benefits of monitoring outweigh the costs of monitoring. In addition, institutional investors will pay attention to the environmental performance of the enterprises. Institutional investors investing in companies with better environmental performance can bring higher stock returns, improve the long-term value of enterprises [7], and affect the sustainable development performance of enterprises. Accordingly, this paper proposes Hypothesis 2:

**Hypothesis 2 (H2).** *The higher the shareholding ratio of institutional investors, the better the sustainable development performance of the company.*

Media attention and institutional investors, as the external supervision mechanism of enterprises, can effectively reduce information asymmetry and affect the decision-making of enterprise operation and management, thus safeguarding the interests of stakeholders. To sum up, when the shareholding ratio of institutional investors is high, the more contact institutional investors have with the management of the company, the more significant the information advantage they have. At the same time, through information integration, institutional investors have increased or decreased their holdings of companies in a timely manner to convey to the outside world the "signal" of good or bad business conditions, reducing the need for media reports to obtain information and reducing the effectiveness of the media as an intermediary in obtaining information. When the shareholding ratio of institutional investors is small, institutional investors have less contact and limited access to information with enterprises, and are more inclined to refer to relevant content reported by the media when making decisions. At this point, media reports enhance the transparency of the external information environment, save the investors' information costs, and replace the role played by institutional investors to a certain extent. Accordingly, this paper proposes research Hypothesis 3:

**Hypothesis 3 (H3).** *There is a substitution effect between media attention and institutional investor holdings.*

### 3.2. The Adjustment Effect of the Level of Internal Control of the Enterprise

As the external supervision mechanism of enterprises, media attention and institutional investor shareholding will have a certain impact on the decision-making of enterprise operation and management. However, whether the final decision of an enterprise meets the

requirements of stakeholders also depends on the level of internal control of the enterprise. Enterprises with a high level of internal control will accurately and effectively transmit the information transmitted by the media and institutional investors to enterprise managers, avoiding the risk of adverse selection by fewer managers for their own interests, and reducing the risk of enterprise losses. In addition, the media and institutional investors jointly play the role of external supervision, which can also improve the internal control level of enterprises and promote the development of enterprises. Under the adjustment effect of the expected performance gap, the media and institutional investors urge enterprises to improve the level of internal control and help enterprises make business decisions in line with the stakeholders, and participate in corporate governance [49].

In contrast, companies with lower levels of internal control are at increased risk of making adverse choices focused on their own interests. At the same time, the supervision effect of the media and institutional shareholders is weakened, and enterprises cannot identify external information transmitted by the media and institutional investors in a timely manner, thus making business decisions that are detrimental to stakeholders and the sustainable development of the company. Accordingly, this paper proposes Hypothesis 4:

**Hypothesis 4 (H4).** *The level of corporate internal control can moderate the impact of media attention on the institutional investors' shareholding on corporate sustainability performance.*

Based on the above analysis, this paper constructed a conceptual framework for media attention, institutional investor shareholding, and corporate sustainability performance, as shown in Figure 1.

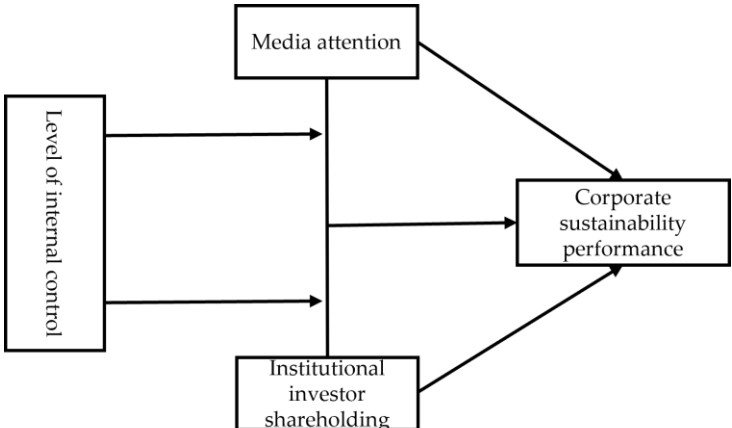

**Figure 1.** Conceptual framework for the effect of the level of internal control moderating media attention and institutional investor shareholding on the corporate sustainability performance.

## 4. Research Design

### 4.1. Sample Selection and Data Sources

This paper selected China's A—share listed companies from 2010 to 2020 as the initial sample for the study excluding the financial industry, ST/PT or delisting, and missing data samples, and a total of 829 observation samples were obtained. The media attention in this paper was obtained from the CNRDS database, and other data were from the WIND and CSMAR databases.

### 4.2. Variable Selection

#### 4.2.1. Explained Variable

Corporate Sustainability Performance (Sustain). Referring to the research of Xie and Zhu [50], this paper divided corporate sustainability performance into two dimensions: financial performance and environmental performance. Among them, the financial performance of enterprises is measured by the return on total assets (ROA), and the environmental

performance is measured by the environmental score of corporate social responsibility listed on Hexun.com as a proxy variable of environmental performance. The entropy weight method can give weights to the relevant rating indicators on the basis of continuous changes, and reasonably and objectively reflect the importance of the selected indicators in the entire indicator system. Therefore, this paper refers to the research of Xi and Zhao [51], and the selection of the entropy weight method to measure the sustainable development performance of enterprises by the index calculated by the comprehensive calculation of indicators has certain scientific and reasonable characteristics.

### 4.2.2. Explanatory Variables

Media attention (Media). Media attention refers to the media to convey relevant information to the market and stakeholders, with the help of public opinion pressure to play a certain role in the supervision, restraint, or guidance of the behavior of the company and stakeholders. In terms of the measurement of media attention, most of the research is based on traditional media such as newspapers and periodicals, but with the development of the Internet, online media has increasingly become an important medium for people to acquire the information of listed companies. In light of this, this paper referred to the research of Yuan and Xiong [14] and Wang and Liao [52] to further select the number of reports by newspapers and online media on enterprises plus 1 to use the logarithm to measure the media attention received by enterprises.

Institutional Investor Holding (Inst). Referring to the research of Wang et al. [53], this paper took the sum of the shareholding ratios of several types of institutional investors such as securities investment funds, social security funds, securities companies, insurance companies, and trust companies as a proxy variable for the institutional investors' shareholding.

### 4.2.3. Adjustment Variables

Internal Control (IC). This paper used the internal control index of Chinese listed companies provided by Dibo, China's internal control index database, and took its natural logarithm as the internal control indicator (IC). The internal control index provided by Shenzhen Dibo Company in China can comprehensively reflect the company's internal control level and risk management control level. It has high authority and can objectively reflect the company's internal control level.

### 4.2.4. Control Variables

According to the relevant research of listed companies. This paper selected company size (Size), asset-liability ratio (Lev), corporate growth (Growth), number of directors (Board), proportion of independent directors (Indep), two-in-one (Dual), ownership concentration (Top1), and shareholder balance (Balance) as the control variable in this paper. Table 1 shows a description of the variables.

**Table 1.** Variable description.

| Variable Category | Variable Name | Variable Name | Variable Definitions |
|---|---|---|---|
| Explained variable | Sustainability Performance | Sustain | According to the return on total assets (ROA) of the company and the social and environmental responsibility score of Hexun.com, it is calculated using the entropy weight method. |
| Explanatory variables | Media attention | Media | Add 1 to the number of newspapers and Internet business information reports to get the logarithmic value |
| | Institutional investor shareholding | Inst | Institutional investors' share of total equity |
| Moderator | Level of internal control | IC | Add one to the internal control index provided by Dibo Company to get the logarithmic value |
| Control variables | Company Size | Size | Natural logarithm of annual total assets |
| | Assets and liabilities | Lev | Annual Total Liabilities/Annual Total Assets |
| | Business growth | Growth | Operating income for the current year/operating income for the previous year−1 |
| | Number of directors | Board | The natural logarithm of the number of directors |

| Variable Category | Variable Name | Variable Name | Variable Definitions |
|---|---|---|---|
| | Proportion of independent directors | Indep | Number of independent directors/directors |
| | Two jobs in one | Dual | 1 if the chairman and manager are the same person, 0 otherwise |
| | Ownership concentration | Top1 | Number of shares held by the largest shareholder/total number of shares |
| | Equity balance | Balance | The sum of the shareholding ratio of the second to five largest shareholders/the shareholding ratio of the first largest shareholder |

### 4.3. Model Construction

In order to examine the impact of the media and institutional investors' attention on corporate sustainability performance, this paper constructed models (1) and (2).

$$\text{Sustain} = \alpha_0 + \alpha_1 \text{Media} + \sum_{i=2}^{9} \text{controls} + \epsilon \tag{1}$$

$$\text{Sustain} = \alpha_0 + \alpha_1 \text{Inst} + \sum_{i=2}^{9} \text{controls} + \epsilon \tag{2}$$

Among them, Media represents the media attention, Inst represents the institutional investment shareholding ratio, Controls represents all of the control variables, and $\epsilon$ represents the random disturbance term. If $\alpha_1$ in model (1) is significantly positive, then H1 is established; similarly, if $\alpha_1$ in model (2) is significantly positive, then H2 is established.

In order to examine the possible substitution effect between media attention and institutional investors' shareholding, this paper built model (3) with reference to the practice of Chen et al. [49], and introduced the interaction term between media attention and institutional investors' shareholding ratio into the equation. For example, if $\alpha_3$ is significantly negative, it indicates that there is a substitution effect between the two, that is, when media attention and institutional investor shareholding coexist, the overall improvement effect on the corporate sustainable development performance decreases, that is, H3 is proven.

$$\text{Sustain} = \alpha_0 + \alpha_1 \text{Media} + \alpha_2 \text{Inst} + \alpha_3 \text{Media} \times \text{Inst} + \sum_{i=3}^{10} \text{controls} + \epsilon \tag{3}$$

The three-dimensional interaction term Media × Inst × IC is further introduced to construct model (4) to explore the level of internal control that should have a moderating effect on the substitution effect of media attention and institutional investor shareholding in the process of improving the performance of corporate sustainable development. If the two-dimensional interaction item Media × Inst coefficient $\alpha_4$ is still significant and the three-dimensional interaction item Media × Inst × IC is significantly positive, it indicates that the improvement in the internal control level of the enterprise can significantly weaken the substitution effect between the two, and H4 is verified.

$$\text{Sustain} = \alpha_0 + \alpha_1 \text{Media} + \alpha_2 \text{Inst} + \alpha_3 \text{IC} + \alpha_4 \text{Media} \times \text{Inst} + \alpha_5 \text{Media} \times \text{IC} + \alpha_6 \text{Inst} \times \text{IC} + \alpha_7 \text{Media} \times \text{Inst} \times \text{IC} + \sum_{i=8}^{15} \text{controls} + \epsilon \tag{4}$$

## 5. Empirical Analysis and Discussion

### 5.1. Descriptive Statistics

The descriptive statistics of the main variables in this paper are shown in Table 2. The average value of the enterprise sustainable development performance in Table 2 was 0.524, the standard deviation was 0.518, the minimum value was 0.013, and the maximum value

was 0.851, indicating that there were certain differences in the sustainable development performance of the sample enterprises, and that the level of the sustainable development performance of enterprises needs to be further improved. The average of media attention was 4.389, the standard deviation was 1.212, the minimum value was 0, and the maximum value was 8.04, the media attention level was in an imbalance, and some companies were less reported by the media. The average value of the institutional investors' shareholding was 0.520, the standard deviation was 0.197, and the minimum and maximum values were 0 and 0.987, respectively, indicating that there was a large difference in the shareholding ratio of institutional investors between enterprises, and some enterprises lacked institutional investment. The average value of the internal control level of the enterprise was 6.5, the standard deviation was 0.15, and the minimum and maximum values were 2.194 and 6.903, respectively, and the internal control level of the enterprise showed individual differences.

**Table 2.** Descriptive statistics results.

| Variable | Obs | Mean | Std. Dev. | Min | Max |
|---|---|---|---|---|---|
| Sustain | 9117 | 0.524 | 0.518 | 0.0130 | 0.851 |
| Media | 9119 | 4.389 | 1.212 | 0 | 8.04 |
| Inst | 9119 | 0.520 | 0.197 | 0 | 0.987 |
| IC | 9023 | 6.5 | 0.15 | 2.194 | 6.903 |
| Size | 9119 | 22.723 | 1.39 | 18.833 | 28.636 |
| Lev | 9119 | 0.492 | 0.194 | 0.022 | 1.352 |
| Growth | 9119 | 0.447 | 19.888 | −0.953 | 1878.372 |
| Board | 9106 | 2.176 | 0.2 | 1.386 | 2.996 |
| Indep | 9106 | 0.372 | 0.057 | 0.125 | 0.8 |
| Dual | 8945 | 0.153 | 0.36 | 0 | 1 |
| Top 1 | 9119 | 0.356 | 0.156 | 0.034 | 0.9 |
| Balance | 9119 | 0.576 | 0.567 | 0.004 | 3.731 |

*5.2. Correlation Analysis*

It can be seen from Table 3 that the coefficient of corporate sustainability performance and media attention was 0.057, which is significant at the 1% level, and the coefficient with the institutional investors' shareholding ratio was 0.167, which is significant at the 1% level. It indicates that the greater the media attention and the shareholding ratio of institutional investors, the better the sustainable development performance of the enterprise, which initially verifies hypotheses H1 and H2.

**Table 3.** Correlation test.

| | Sustain | Media | Inst | Inc | Size | Lev | Growth | Board | Indep | Dual | Top 1 | Balance |
|---|---|---|---|---|---|---|---|---|---|---|---|---|
| Sustain | 1.000 *** | | | | | | | | | | | |
| media | 0.057 *** | 1.000 *** | | | | | | | | | | |
| inst | 0.167 *** | 0.064 *** | 1.000 *** | | | | | | | | | |
| inc | 0.309 *** | 0.061 *** | 0.193 *** | 1.000 *** | | | | | | | | |
| Size | −0.002 | 0.225 *** | 0.432 *** | 0.205 *** | 1.000 *** | | | | | | | |
| Lev | −0.337 *** | 0.142 *** | 0.148 *** | 0.041 *** | 0.464 *** | 1.000 *** | | | | | | |
| Growth | −0.001 | 0.013 | 0.032 *** | 0.020 * | 0.037 *** | 0.025 ** | 1.000 *** | | | | | |
| Board | 0.014 | 0.030 *** | 0.199 *** | 0.075 *** | 0.223 *** | 0.093 *** | 0.027 ** | 1.000 *** | | | | |
| Indep | −0.031 *** | 0.033 *** | −0.031 *** | 0.021 ** | 0.085 *** | 0.041 *** | −0.008 | −0.418 *** | 1.000 *** | | | |
| Dual | 0.015 | 0.035 *** | −0.182 *** | −0.043 *** | −0.064 *** | −0.047 *** | 0.024 ** | −0.160 *** | 0.084 *** | 1.000 *** | | |
| Top 1 | 0.080 *** | −0.037 *** | 0.634 *** | 0.121 *** | 0.233 *** | 0.100 *** | −0.001 | 0.052 *** | 0.043 *** | −0.132 *** | 1.000 *** | |
| Balance | 0.018 * | 0.079 *** | −0.107 *** | −0.022 ** | 0.032 *** | −0.044 *** | 0.027 ** | 0.053 *** | −0.051 *** | 0.082 *** | −0.643 *** | 1.000 *** |

Note: ***, ** and * represent the significance levels of 1%, 5%, and 10%, respectively, the same below.

### 5.3. Analysis of Regression Results

In order to verify the above hypothesis, the regression results are shown in Table 4. Result (2) showed that media attention and corporate sustainability performance were significantly positively correlated at the level of 1%, indicating that the more media attention, the better and more transparent the information, and the more conducive it is to the development of the economic and environmental benefits; therefore, Hypothesis H1 is established. Result (3) showed that the sustainable development performance of enterprises was significantly positively correlated with the shareholding ratio of institutional investors at the level of 1%, indicating that the higher the shareholding ratio of the institutional investors, the more conducive it is to give full play to their capital and information advantages to supervise and constrain enterprises, and to improve the sustainable development performance of enterprises, Hypothesis H2 is established.

**Table 4.** Analysis of the regression results.

| | (1) Sustain | (2) Sustain | (3) Sustain | (4) Sustain | (5) Sustain |
|---|---|---|---|---|---|
| Media | | 0.002 *** | | 0.004 *** | 0.108 *** |
| | | (4.77) | | (4.10) | (3.11) |
| Inst | | | 0.048 *** | 0.068 *** | 1.094 *** |
| | | | (9.69) | (6.98) | (3.36) |
| Media × Inst | | | | −0.005 ** | −0.225 *** |
| | | | | (−2.47) | (−3.46) |
| IC | | | | | 0.131 *** |
| | | | | | (4.93) |
| Media × IC | | | | | −0.016 *** |
| | | | | | (−2.96) |
| Inst × IC | | | | | −0.159 *** |
| | | | | | (−3.17) |
| Media × Inst × IC | | | | | 0.034 *** |
| | | | | | (3.38) |
| Size | 0.012 *** | 0.012 *** | 0.010 *** | 0.010 *** | 0.008 *** |
| | (12.95) | (12.70) | (10.57) | (10.16) | (8.25) |
| Lev | −0.113 *** | −0.113 *** | −0.109 *** | −0.109 *** | −0.098 *** |
| | (−28.81) | (−28.86) | (−27.95) | (−27.99) | (−25.85) |
| Growth | 0.000 | 0.000 | 0.000 | 0.000 | −0.000 |
| | (0.39) | (0.33) | (0.14) | (0.24) | (−0.28) |
| Board | −0.003 | −0.002 | −0.003 | −0.002 | −0.003 |
| | (−0.68) | (−0.55) | (−0.79) | (−0.63) | (−0.88) |
| Indep | 0.001 | 0.000 | 0.001 | 0.000 | 0.000 |
| | (0.08) | (0.04) | (0.09) | (0.02) | (0.01) |
| Dual | −0.001 | −0.001 | −0.001 | −0.001 | −0.001 |
| | (−0.95) | (−0.97) | (−0.93) | (−0.92) | (−0.48) |
| Top1 | 0.034 *** | 0.034 *** | −0.012 | −0.012 | −0.014 * |
| | (5.11) | (5.09) | (−1.52) | (−1.45) | (−1.75) |
| Dturn | 0.000 | −0.001 | 0.001 | −0.001 | −0.001 |
| | (0.33) | (−0.53) | (0.56) | (−0.49) | (−1.11) |
| Balance | 0.004 *** | 0.004 ** | −0.005 *** | −0.005 *** | −0.005 *** |
| | (2.69) | (2.54) | (−2.79) | (−2.89) | (−3.18) |
| _cons | 0.306 *** | 0.301 *** | 0.348 *** | 0.336 *** | −0.470 *** |
| | (11.23) | (11.06) | (12.69) | (12.22) | (−2.71) |
| Year | | | Yes | | |
| Industry | | | Yes | | |
| Adj $R^2$ | 0.044 | 0.047 | 0.055 | 0.058 | 0.111 |

Note: ***, ** and * represent the significance levels of 1%, 5%, and 10%, respectively.

From Result (4), the coefficient of the interaction item Media × Inst was −0.005, which was significant at the level of 1%, indicating that on one hand, the increase in institu-

tional investors' shareholding weakened the effect of media attention on improving the performance of corporate sustainable development. On the other hand, the increased media attention will also weaken the supervisory role played by institutional investors to a certain extent. When the two coexist, the overall improvement effect on the sustainable development performance of enterprises will decrease. Hypothesis 3 was established. The coefficient of the three-dimensional interaction term Media × Inst × IC for the level of internal control of media attention and institutional investors' shareholding was significantly and positively related to corporate sustainability performance at the 1% level. This indicates that with the improvement in the internal control level of enterprises, it is easier for enterprises to make business decisions that are in line with the interests of stakeholders, the risk of adverse selection is reduced, and the supervision role of the media and institutional investors is strengthened, which is conducive to the sustainable development performance of enterprises. Thus, Hypothesis H4 is established.

### 5.4. Robustness Test
#### 5.4.1. Endogenous Test

Considering that there may be an endogenous problem between media attention and institutional investors' shareholding on the sustainable development of enterprises, this paper referred to Wang and Liao [52] to select the average value of media attention in the same industry in the same year as the instrumental variable of media attention (M-Media), referring to Wang and Guo [54], the average value of the institutional investors' shareholding by industry and year was selected as the instrumental variable of institutional investor's shareholding (M-Inst), and the two-stage least squares method (2SLS) was adopted. The endogenous regression test was performed, and the regression results are shown in Table 5. For the test of the reasonableness of the instrumental variable selection, the Kleibergen-Paap rk LM statistics of the unidentifiable test were 119.07 and 30.88, respectively, when the instrumental variables M-Media and M-Inst were regressed to media attention and institutional investor holdings in the first stage. All were significant at the 1% level, rejecting the original hypothesis of "insufficient identification of tool variables". In addition, the values of the Kleibergen-Paap rk Wald F statistic in the weak instrumental variable test were 193.04 and 32.68, respectively. Both were much larger than the critical value of the Stock–Yogo weak recognition test at the 10% level of 16.38, rejecting the null hypothesis that "there are weak instrumental variables". Over-identification tests were all just-identified. In summary, the tool variables selected in this article are valid. According to the regression results (1) and (3) in the first stage, the coefficients of media attention and the institutional investors' instrument variables were significantly positive at the 5% and 1% levels, respectively; in the second stage, the coefficients of media attention and the institutional investors' shareholding ratio were 0.005 and 0.241 respectively, which were significant at the level of 1%. Both media attention and institutional investor shareholding significantly improved the level of corporate sustainable development performance, which is consistent with the conclusions of the main tests H1 and H2.

**Table 5.** Two-stage least squares (2SLS) regression results.

| Variables | (1)<br>First<br>Media | (2)<br>Second<br>Sustain | (3)<br>First<br>Inst | (4)<br>Second<br>Sustain |
| --- | --- | --- | --- | --- |
| M-Media | 0.926 ***<br>(13.78) | | | |
| M-Inst | | | 0.441 ***<br>(5.87) | |
| Media | | 0.005 **<br>(2.01) | | |
| Inst | | | | 0.241 ***<br>(3.94) |

**Table 5.** *Cont.*

| Variables | (1) First Media | (2) Second Sustain | | (3) First Inst | (4) Second Sustain |
|---|---|---|---|---|---|
| Constant | −5.303 *** | 0.415 *** | | −0.967 *** | 0.588 *** |
| | (−14.22) | (42.04) | | (−20.25) | (12.48) |
| Controls | | | Yes | | |
| Year | | | Yes | | |
| Industry | | | Yes | | |
| Adj R² | 0.28 | | | 0.63 | |

Note: ***, ** and * represent the significance levels of 1%, 5%, and 10%, respectively.

### 5.4.2. Robustness Check

In order to verify the credibility of the empirical results, this paper conducted the following robustness tests. ① The main explanatory variables, media attention, and institutional investor shareholding were abbreviated by 1%, the results were (1) and (2) in Table 6, and the significance and sign of the coefficients did not change significantly. ② In order to alleviate the possible problem of mutual causality, this paper further used the media attention and institutional investor stockholdings of one lag period as new explanatory variables, and the regression results (3) and (4) in Table 6 show that the significance level and coefficient sign are consistent with the main test. ③ In order to alleviate the endogenous problem caused by missing variables, this paper added the enterprise value (TobinQ) as a new control variable, the regression results are (5) and (6) in Table 6 and the findings remained unaffected when the new control variables were added.

**Table 6.** Robustness test.

| | (1) Sustain | (2) Sustain | (3) Sustain | (4) Sustain | (5) Sustain | (6) Sustain |
|---|---|---|---|---|---|---|
| Media_w | 0.002 *** | | | | | |
| | (4.75) | | | | | |
| Inst_w | | 0.049 *** | | | | |
| | | (9.68) | | | | |
| L.Media | | | 0.004 *** | | | |
| | | | (2.80) | | | |
| L.Inst | | | | 0.006 *** | | |
| | | | | (6.23) | | |
| Media | | | | | 0.001 ** | |
| | | | | | (2.19) | |
| Inst | | | | | | 0.031 *** |
| | | | | | | (6.17) |
| TobinQ | | | | | 0.006 *** | 0.006 *** |
| | | | | | (16.23) | (15.11) |
| _cons | 0.301 *** | 0.348 *** | 0.303 *** | 0.335 *** | 0.223 *** | 0.257 *** |
| | (11.06) | (12.68) | (11.05) | (12.14) | (8.10) | (9.19) |
| Year | | | | Yes | | |
| Industry | | | | Yes | | |
| Adj R² | 0.047 | 0.055 | 0.043 | 0.049 | 0.075 | 0.079 |

Note: ***, ** and * represent the significance levels of 1%, 5%, and 10%, respectively.

## 6. Research Conclusions and Implications

Taking Chinese listed companies as a sample, this paper discussed the impact of media attention and institutional investor shareholding on the sustainable development performance of the listed companies as well as the interaction of the two at the same time. Both can significantly improve the sustainable development performance of the listed companies; the interaction between media attention and institutional investor shareholding will have a reverse effect on the sustainable development performance of enterprises, and

the two have significant alternatives in affecting the sustainable development performance of enterprises effect. However, under the adjustment of the internal control level of the enterprise, the synergistic relationship between the media and institutional investors is finally strengthened.

In China, where the market supervision mechanism is not perfect, the media, as an external supervisor, can, to a certain extent, regulate the internal governance of enterprises through news reports, influence the behavior of management, and play an important role in improving the sustainable development performance of enterprises. At the same time, institutional investors, with their capital and information advantages, tend to be rational in their investment decisions, pay more attention to the long-term development of enterprises, and monitor enterprises based on their own interests At the same time, institutional investors, with their capital and information advantages, tend to make rational investment decisions, pay more attention to the long-term development of enterprises, monitor listed enterprises based on their own interests, and promote the long-term development of enterprises. Therefore, in order to improve the sustainability of enterprises, this paper proposed the following policy insights for enterprises, media, institutional stakeholders, and government, respectively.

For enterprises, they should optimize the equity structure and increase the share of equity held by institutional investors. On one hand, enterprises should vigorously develop institutional investors and encourage them to play a supervisory and governance role and actively participate in corporate governance. Institutional investors have investment advantages such as investment experience and investment strength compared to small- and medium-sized investors, and enterprises as investees should actively introduce different institutional investors to contribute to the development of the enterprises themselves. On the other hand, enterprises should improve the internal control system, promote the reform and improvement of the internal control system of listed enterprises in depth, and restrain the operation managers through the improvement in the internal control system to obtain objective and correct operation information and make correct decisions. The sustainable development of an enterprise requires a good internal control system as a support, and the management of the company should effectively recognize the important role played by internal control in corporate governance, improve corporate information and communication mechanisms, and obtain internal management information related to the company in a timely manner so that it can be accurately and smoothly transmitted between inside and outside the company, which ensures effective communication. Enterprises should strengthen internal supervision, monitor and test the whole process of internal control implementation, assess the quality of internal control, and improve internal control deficiencies in a timely manner. In addition, corporate managers should be aware of the impact of collaborative internal and external governance on sustainable development, and actively guide the media and the public to pay moderate attention to the company, so that incentives and constraints are compatible, and internal and external governance is collaborative to promote the sustainable development of the company.

For the media, the media should pay attention to the sustainable development ability of enterprises in order to regulate their business management. At the same time, it should ensure the truthfulness and accuracy of its reporting content and effectively play its external supervision role. On one hand, the media should disseminate truthful, reliable and accurate information for the public as much as possible to maintain the image and credibility of the media itself. On the other hand, traditional media and emerging media should complement each other, promote each other, and integrate with each other. Specifically, traditional media should keep pace with the times and take the initiative to use the advantages of information technology to launch online, and electronic versions to improve the timeliness and interactivity of information. Emerging media can extend the capacity and depth of media through information technology, enhance the professionalism, and comprehensiveness of information, so that the public and stakeholders can have more

understanding of enterprises, which can regulate the behavior of managers and promote the sustainable development of enterprises.

For institutional shareholders, on one hand, institutional shareholders should improve their willingness to interact and communicate with listed companies, make full use of their advantages, and exert their monitoring rights. In particular, institutional shareholders should give full play to their ability to obtain, analyze, interpret, and disseminate information to better form an effective external restraining force and regulate corporate behavior. On the other hand, institutional shareholders should establish the correct investment consciousness, pay attention to corporate environmental performance, remove enterprises with very poor corporate environmental performance from their investment portfolios, and give priority to investing in enterprises with good environmental performance to force enterprises to pay attention to corporate environmental performance and promote their sustainable development through the feedback mechanism of the capital market.

Regarding the government, it should pay attention to the role of the media and institutional investors in information transmission and public opinion supervision. In addition to the restraint of laws and regulations, more consideration should be given to the restraining role of external non-government sectors such as media supervision and institutional investor supervision to compensate for the inefficiency brought by simple government and legal regulation.

The limitations of this paper are as follows. First, this paper did not distinguish between the nature of the media coverage (positive, negative, and neutral) and the level of media coverage (CCTV media, provincial media, etc.). Although media of different natures (or levels) can effectively monitor listed companies by tracking corporate information, the monitoring effect of media coverage of different nature (or level) may be different. Therefore, a follow-up study can be conducted on the impact of different types of media coverage on the sustainable development of enterprises. Second, there is heterogeneity among institutional investors. Different types of institutional investors have different investment strategies, management models, and capital scales, and therefore their corporate governance capabilities and willingness differ, so future research could consider the impact of institutional investor heterogeneity on corporate sustainability.

**Author Contributions:** All authors contributed to this article. C.L. designed and wrote the paper, and X.W. was responsible for organizing the research and analyzing the data. All authors have read and agreed to the published version of the manuscript.

**Funding:** This research received no external funding.

**Institutional Review Board Statement:** Not applicable.

**Informed Consent Statement:** Not applicable.

**Data Availability Statement:** The data presented in this study are available on request from the corresponding author.

**Conflicts of Interest:** The authors declare no conflict of interest.

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
