# Peer review of "Media and Institutional Investors Focus on the Impact on Corporate Sustainability Performance"

_sustainability, doi:10.3390/su142113878_

Round 1
Reviewer 1 Report
Thank you for offering me the opportunity to review this paper. This paper studies an interesting topic - the impact of media attention and institutional investor attention, and their interaction on the sustainable development performance of enterprises. The authors collected data from China's A-share listed companies from 2010 to 2020. I very much appreciate the authors’ research efforts in this research area. My suggestions are as follows:
1. The definition of corporate sustainability performance. The authors roughly define corporate sustainability performance in the paper – “this paper uses the index calculated by the entropy weight method to measure the performance level of enterprise sustainable development.” It is unclear to readers how the entropy weight method works. Also, the authors should provide references in the paper to support the legitimacy of the method.
2. The definition of media attention. The authors did not provide any reference to the definition of media attention. In the paper, the authors state – “This study uses the number of annual media reports of enterprises to measure the media attention. The specific method is to select the corporate information reported by traditional media such as newspapers and periodicals of various listed companies and the Internet. Simply using the number of annual media reports of enterprises seems like a stretch. The authors should provide references to it.
3. The conceptual framework. The authors may consider adding a figure to summarize the paper's conceptual framework. This will help the readers understand the paper much better.
4. Endogeneity. The authors need to provide rational and empirical tests to show why the endogeneity correction works.
I hope the authors find the above suggestions useful.
Author Response
Thank you for your valuable comments. We have revised the manuscript. Please refer to the attachment for details.

Reviewer 2 Report
The references focus on the Chinese literature. The authors should also review non-Chinese relevant papers.
Author Response

(The authors gave the same response as above.)

Reviewer 3 Report
In general, the paper is presented well. However, some issues need to be fixed and improved.
Firstly in line 10 the sample of the companies is determined from 2010-2015, but in lines 51-52, 119 and in other lines the sample is determined from 2010 to 2020. The dates must be corrected.
Moreover, Further Research and Limitations must added at the end of the article.
In addition more References must be reported to the Literature Review.
Also, in Keywords (lines 21 to 22) the word Internal Control Indicators as well as the Abbreviation of Internal Control (IC) must be added next to the mentioned word.
I hope that the authors will find the comments constructive and consider these as an attempt to help them enhance the quality of their manuscript.
Author Response

(The authors gave the same response as above.)

Reviewer 4 Report
The topic addressed is interesting for current research and we believe that the empirical study really contributes to the development of knowledge in the field of analysis the factors with significant impact on corporate sustainability performance. The chosen topic is catchy and the empirical study is original. However, for a thorough examination of the correlations and anchoring the subject in the current state of knowledge we recommend the following to the authors:
1. The introduction does not address research questions. Add a least one to make the readers interested in your work.
2. Reconsider section 2.1 and 2.2, adding a few more empirical works that tackle the same topic. This section should be much richer in citations to anchor the study in the existing current knowledge.
3. A figure of your theoretical research framework it would add value to your work.
4. Section 5.1. misses an explanation of the descriptive statistics analysis. It can’t stay that way. Insert a description.
5. Highlight your contributions more. It remains unclearly presented in the paper which are strenghts and weaknesses compared to other studies on the topic.
6. Revise the Conclusion section to make clear to the readers which are the governance and managerial implications of your work for the listed companies as well as the implications at the level of institutional stakeholders. Present also the limits of the research.
7. Polish the grammar and check the spelling in each section of the manuscript
Author Response

(The authors gave the same response as above.)

Round 2
Reviewer 1 Report
I recommend accepting the paper.
Reviewer 4 Report
All suggested and recommended changes were made. Good luck with your research publication!